# Are Guessing, Source Coding and Tasks Partitioning Birds of A Feather? [note 1]

**DOI:** 10.3390/e24111695

**Published:** 2022-11-19

**Authors:** M. Ashok Kumar, Albert Sunny, Ashish Thakre, Ashisha Kumar, G. Dinesh Manohar

**Affiliations:** 1Department of Mathematics, Indian Institute of Technology Palakkad, Palakkad 678557, India; 2Department of Computer Science and Engineering, Indian Institute of Technology Palakkad, Palakkad 678557, India; 3Department of Mathematics, Indian Institute of Technology Indore, Indore 453552, India; 4Institute for Communications Engineering, Technical University of Munich, 80333 Munich, Germany

**Keywords:** guessing, source coding, tasks partitioning, Shannon entropy, Rényi entropy, Kullback–Leibler divergence, relative α-entropy, Sundaresan’s divergence, 94A15, 94A17, 94A50, 62B10

## Abstract

This paper establishes a close relationship among the four information theoretic problems, namely Campbell source coding, Arikan guessing, Huleihel et al. memoryless guessing and Bunte and Lapidoth tasks’ partitioning problems in the IID-lossless case. We first show that the aforementioned problems are mathematically related via a general moment minimization problem whose optimum solution is given in terms of Renyi entropy. We then propose a general framework for the mismatched version of these problems and establish all the asymptotic results using this framework. The unified framework further enables us to study a variant of Bunte–Lapidoth’s tasks partitioning problem which is practically more appealing. In addition, this variant turns out to be a generalization of Arıkan’s guessing problem. Finally, with the help of this general framework, we establish an equivalence among all these problems, in the sense that, knowing an asymptotically optimal solution in one problem helps us find the same in all other problems.

## 1. Introduction

The concept of entropy is very central to information theory. In source coding, the expected number of bits required (per letter) to encode a source with (finite) alphabet set X and probability distribution *P* is the Shannon entropy $H\left(P\right):=-\sum_{x\in X}P\left(x\right)\log P\left(x\right)$. If the compressor does not know the true distribution *P*, but assumes a distribution *Q* (mismatch), then the number of bits required for compression is H(P)+I(P,Q), where
(1)$$I\left(P,Q\right):=-\sum_{x\in X}P\left(x\right)\log Q\left(x\right)+\sum_{x\in X}P\left(x\right)\log P\left(x\right),$$
is the entropy of *P* relative to *Q* (or the Kullback–Leibler divergence). In his seminal paper, Shannon [1] argued that H(P) can also be regarded as a measure of uncertainty. Subsequently, Rényi [2] introduced an alternate measure of uncertainty, now known as *Rényi entropy* of order α, as
(2)$$\begin{array}{c} H_{\alpha }\left(P\right):=\frac{1}{1-\alpha }\log\sum_{x\in X}P{\left(x\right)}^{\alpha }, \end{array}$$
where α>0 and α≠1. Rényi entropy can also be regarded as a generalization of the Shannon entropy as $\lim_{\alpha \to 1}H_{\alpha }\left(P\right)=H\left(P\right)$. Refer Aczel and Daroczy [3] and the references therein for an extensive study of various measures of uncertainty and their characterizations.

In 1965, Campbell [4] gave an operational meaning to Rényi entropy. He showed that, instead of expected code lengths, if one minimizes the cumulants of code lengths, then the optimal cumulant is Rényi entropy $H_{\alpha }\left(P\right)$, where α=1/(1+ρ) with ρ being the order of the cumulant. He also showed that the optimal cumulant can be achieved by encoding sufficiently long sequences of symbols. Sundaresan (Theorem 8 of [5]) (c.f. Blumer and McElice [6]) showed that, in the mismatched case, the optimal cumulant is $H_{\alpha }\left(P\right)+I_{\alpha }\left(P,Q\right)$, where
(3)$$I_{\alpha }\left(P,Q\right):=\frac{\alpha }{1-\alpha }\log\sum_{x}P\left(x\right){\left(\frac{Q{\left(x\right)}^{\alpha }}{\sum_{y}Q{\left(y\right)}^{\alpha }}\right)}^{\frac{\alpha -1}{\alpha }}-H_{\alpha }\left(P\right)$$
is called *α-entropy of P relative to Q* or *Sundaresan’s divergence* [7]. Hence, $I_{\alpha }\left(P,Q\right)$ can be interpreted as the penalty for not knowing the true distribution. The first term in (Equation 3) is sometimes called the Renyi cross-entropy and is analogous to the first term of (Equation 1). $I_{\alpha }\left(P,Q\right)\geq 0$ with equality if and only if P=Q. $I_{\alpha }$-divergence can also be regarded as a generalization of the Kullback–Leibler divergence as $\lim_{\alpha \to 1}I_{\alpha }\left(P,Q\right)=I\left(P,Q\right)$. Refer to [5,8,9] for detailed discussions on the properties of $I_{\alpha }$. Lutwak et al. also independently identified $I_{\alpha }$ in the context of maximum Rényi entropy and called it an *α-Rényi relative entropy* (Equation (Equation 4) of [10]). $I_{\alpha }$, for α>1, also arises in robust inference problems (see [11] and the references therein).

In [12], Massey studied a guessing problem where one is interested in the expected number of guesses required to guess a random variable *X* that assumes values from an infinite set, and found a lower bound in terms of Shannon entropy. Arıkan [13] studied it for a finite alphabet set and showed that Rényi entropy arises as the optimal solution in minimizing moments of the number of guesses. Subsequently, Sundaresan [5] showed that the penalty in guessing according to a distribution *Q* when the true distribution is *P*, is given by $I_{\alpha }\left(P,Q\right)$. It is interesting to note that guesswork has also been studied from a large deviations point of view [14,15,16,17,18]. Bunte and Lapidoth [8] studied a problem on partitioning of tasks and showed that Rényi entropy and Sundaresan’s divergence play a similar role in the optimal number of tasks performed. We propose, in this paper, a variant of this problem where the tasks in each subset of the partition are performed according to the decreasing order of probabilities. We show that Rényi antropy and Sundaresan’s divergences arise as optimal solutions in this problem too. Huleihel et al. [19,20] studied the memoryless guessing problem, a variant of Arıkan’s guessing problem, with i.i.d. (independent and identically distributed) guesses and showed that the minimum attainable factorial moments of number of guesses is the Rényi entropy. We show, in this paper, that the minimum factorial moment in the mismatched case is measured by the Sundaresan’s divergence.

We observe that, in all these problems, the objective is to minimize usual moments or factorial moments of random variables, and Rényi entropy and Sundaresan’s divergence arise in optimal solutions. The relationship between source coding and guessing is well-known in the literature. Arıkan and Merhav established a close relationship between lossy source coding and guessing with distortion using large deviation techniques [14,21]. The same for the lossless case was done by Hanawal and Sundaresan [17]. In this paper, we establish a general framework for all the five problems in the IID-lossless case. We then use this to establish upper and lower bounds for the mismatched version of these problems. This helps us find an equivalence among all these problems, in the sense that knowing an asymptotically optimal solution in one problem helps us find the same in all other problems.


*Our Contributions in the Paper:*
(a)a general framework for the problems on source coding, guessing and tasks partitioning;(b)lower and upper bounds for the general framework of these problems both in matched and mis-matched cases;(c)a unified approach to derive bounds for the mismatched version of these problems;(d)a generalized tasks partitioning problem; and(e)establishing operational commonality among the problems.



*Organisation of the Paper:*


In Section 2, we present our unified framework, and find conditions under which lower and upper bounds are attained. In Section 3, we present four well-known information-theoretic problems, namely, Campbell’s source coding, Arıkan’s guessing, Huleihel et al.’s memoryless guessing, and Bunte–Lapidoth’s tasks partitioning, and re-establish and refine major results pertaining to these problems. In Section 4, we propose and solve a generalized tasks partitioning problem. In Section 5, we establish a connection among the aforementioned problems. Finally, we summarize and conclude the paper in Section 6.

## 2. A General Minimization Problem

In this section, we present a general minimization problem whose optimum solution evaluates to Rényi entropy. We will later show that all problems stated in Section 3 are particular instances of this general problem.

 **Proposition 1.** 
*Let ψ:X→(0,∞) be such that $\sum_{x\in X}\psi {\left(x\right)}^{-1}\leq k$ for some k>0. Then, for ρ∈(−1,0)∪(0,∞),*

(4)
$$\frac{1}{\rho }\log E_{P}\left[\psi {\left(X\right)}^{\rho }\right]\geq H_{\alpha }\left(P\right)-\log k,$$

*where $E_{P}\left[\cdot \right]$ denotes the expectation with respect to probability distribution P on X, $H_{\alpha }\left(P\right)$ is the Rényi entropy of order α, and α:=α(ρ)=1/(1+ρ). The lower bound is achieved if and only if*

(5)
$$\psi {\left(x\right)}^{-1}=k\cdot P{\left(x\right)}^{\alpha }/Z_{P,\alpha }\;\mathrm{for}\;x\in X,$$

*where $Z_{P,\alpha }:=\sum_{x\in X}P{\left(x\right)}^{\alpha }$.*


 **Proof.** Observe that
$$\begin{array}{ccc} \mathrm{sgn}\left(\rho \right)\sum_{x\in X}P\left(x\right)\psi {\left(x\right)}^{\rho } & = & \mathrm{sgn}\left(\rho \right)\sum_{x\in X}P{\left(x\right)}^{\alpha }{\left(\frac{\psi {\left(x\right)}^{-1}}{P{\left(x\right)}^{\alpha }}\right)}^{-\rho }\\ & \overset{\left(a\right)}{\geq } & \mathrm{sgn}\left(\rho \right)\left(\sum_{x}P{\left(x\right)}^{\alpha }\right)\cdot {\left(\frac{\sum_{x}\psi {\left(x\right)}^{-1}}{\sum_{x}P{\left(x\right)}^{\alpha }}\right)}^{-\rho }\\ & = & \mathrm{sgn}\left(\rho \right){\left(\sum_{x}P{\left(x\right)}^{\alpha }\right)}^{1+\rho }\cdot {\left(\sum_{x}\psi {\left(x\right)}^{-1}\right)}^{-\rho }\\ & \overset{\left(b\right)}{\geq } & \mathrm{sgn}\left(\rho \right){\left(\sum_{x}P{\left(x\right)}^{\alpha }\right)}^{1+\rho }\cdot k^{-\rho }, \end{array}$$
where (a) is due to the generalised log-sum inequality (Equation (4.1) of [22]) applied to the function $f\left(x\right)=\mathrm{sgn}\left(\rho \right)\cdot x^{-\rho }$; and (b) follows from the hypothesis that $\sum_{x}\psi {\left(x\right)}^{-1}$≤k. By taking log and then dividing by ρ, we obtain (Equation 4). Equality holds in (a) if and only if $\psi {\left(x\right)}^{-1}=\nu P{\left(x\right)}^{\alpha }$ for some constant ν and in (b) if and only if $\sum_{x}\psi {\left(x\right)}^{-1}=k$. This completes the proof. □

The left-side of (Equation 4) is called *normalised cumulant of ψ(X) of order ρ*. The measure $P^{\left(\alpha \right)}\left(x\right):=P{\left(x\right)}^{\alpha }/Z_{P,\alpha }$ in (Equation 5) that attains the lower bound in (Equation 4) is called an *α-scaled measure* or *escort measure* of *P*. This measure also arises in robust inference (Equation (Equation 7) of [11]) and statistical physics [23]. The above proposition can also be proved using a variational formula as follows. By a version of Donsker–Varadhan variational formula (Propostion 4.5.1 of [24]), for any real-valued *f* on X, we have
(6)$$\log E_{P}\left[2^{f(X)}\right]=\max_{Q}\left\{E_{Q}\left[f\left(X\right)\right]-D\left(Q∥P\right)\right\},$$
where the max is over all probability distributions *Q* on X. Taking ρ>0 and f(x)=ρlogψ(x) in (Equation 6), we have
$$\begin{array}{ccc} \log E_{P}\left[\psi {\left(X\right)}^{\rho }\right] & = & \max_{Q}\left\{\rho E_{Q}\left[\log\psi \left(X\right)\right]-D\left(Q∥P\right)\right\}\\ & = & \max_{Q}\left\{-\rho \sum_{x}Q\left(x\right)\log\frac{\psi {\left(x\right)}^{-1}Q\left(x\right)}{Q(x)}-D\left(Q∥P\right)\right\}\\ & = & \max_{Q}\left\{\rho H\left(Q\right)-\rho \sum_{x}Q\left(x\right)\log\frac{\psi {\left(x\right)}^{-1}}{Q(x)}-D\left(Q∥P\right)\right\}\\ & \overset{\left(a\right)}{\geq } & \max_{Q}\left\{\rho H\left(Q\right)-\rho \log\sum_{x}\psi {\left(x\right)}^{-1}-D\left(Q∥P\right)\right\}\\ & \overset{\left(b\right)}{\geq } & \rho \max_{Q}\left\{H\left(Q\right)-\frac{1}{\rho }D\left(Q∥P\right)\right\}-\rho \log k, \end{array}$$
where (a) is by the log-sum inequality (Equation (4.1) of [22]) and (b) is by applying the constraint $\sum_{x}\psi {\left(x\right)}^{-1}\leq k$. For ρ∈(−1,0), the inequalities in (a) and (b) are reversed, and the last max is replaced by min. Hence, (Equation 4) follows as the last max is equal to $H_{\alpha }\left(P\right)$ by (Theorem 1 of [25]). Equality in (a) and (b) holds if and only if $\psi {\left(x\right)}^{-1}=k\cdot Q\left(x\right)$. In addition, the last max is attained when $Q\left(x\right)=P{\left(x\right)}^{\alpha }/Z_{P,\alpha }\mathrm{for}x\in X$. This completes the proof. The following is the analogous one for Shannon entropy.

 **Proposition 2.** 
*Let ψ:X→(0,∞) be such that $\sum_{x\in X}\psi {\left(x\right)}^{-1}\leq k$. Then,*

(7)
$$\begin{array}{c} E_{P}\left[\log\psi (X)\right]\geq H\left(P\right)-\log k. \end{array}$$

*Equality in (Equation 7) is achieved if and only if $\psi {\left(x\right)}^{-1}=k\cdot P\left(x\right)\;\;\forall x\in X$.*


**Proof.** $$\begin{array}{ccc} E_{P}\left[\log\psi (X)\right] & = & -\sum_{x}P\left(x\right)\log\frac{P\left(x\right)\cdot \psi {\left(x\right)}^{-1}}{P(x)}=H\left(P\right)-\sum_{x}P\left(x\right)\log\frac{\psi {\left(x\right)}^{-1}}{P(x)}\\ & \geq & H\left(P\right)-\log\sum_{x}\psi {\left(x\right)}^{-1}\geq H\left(P\right)-\log k, \end{array}$$ where the penultimate inequality is due to the log-sum inequality. Equality holds in both inequalities if and only if $\psi {\left(x\right)}^{-1}=k\cdot P\left(x\right)\;\;\forall x\in X$. □

It is interesting to note that $\log E_{P}\left[\psi {\left(X\right)}^{\rho }\right]/\rho \to E_{P}\left[\log\left(\psi (X)\right)\right]$ and $H_{\alpha }\left(P\right)\to H\left(P\right)$ as ρ→0 in (Equation 4). We now extend Propositions 1 and 2 to sequences of random variables. Let $X^{n}$ be the set of all *n*-length sequences of elements of X, and $P_{n}$ be the *n*-fold product distribution of *P* on $X^{n}$, that is, for $x^{n}:=x_{1},\cdots ,x_{n}\in X^{n}$, $P_{n}\left(x^{n}\right)=\prod_{i=1}^{n}P\left(x_{i}\right)$.

 **Corollary 1.** 
*Given any n≥1, if $\psi_{n}:X^{n}\to \left[0,\infty \right)$ is such that $\sum_{x^{n}\in X^{n}}\psi_{n}{\left(x^{n}\right)}^{-1}$ $\leq k_{n}$ for some $k_{n}>0$, then*
(a)
*For ρ∈(−1,0)∪(0,∞),*

$$\underset{n\to \infty }{lim inf}\frac{1}{n\rho }\log E_{P_{n}}\left[\psi_{n}{\left(X^{n}\right)}^{\rho }\right]\geq H_{\alpha }\left(P\right)-\underset{n\to \infty }{lim sup}\frac{\log k_{n}}{n}.$$

(b)

$$\underset{n\to \infty }{lim inf}\frac{1}{n}E_{P_{n}}\left[\log\psi_{n}\left(X^{n}\right)\right]\geq H\left(P\right)-\underset{n\to \infty }{lim sup}\frac{\log k_{n}}{n},$$


*where $E_{P_{n}}\left[\cdot \right]$ denotes the expectation with respect to probability distribution $P_{n}$ on $X^{n}$.*


 **Proof.** It is easy to see that $H_{\alpha }\left(P_{n}\right)=nH_{\alpha }\left(P\right)$ and $H\left(P_{n}\right)=nH\left(P\right)$. Applying Propositions 1 and 2, dividing throughout by *n* and taking lim infn→∞, the results follow. □

### A General Framework for Mismatched Cases

In this sub-section, we establish a unified approach for cases when there is mismatch between assumed and true distributions.

 **Proposition 3.** 
*Let ρ>−1, α=1/(1+ρ), and Q be a probability distribution on X. For n≥1, let $Q_{n}$ be the n-fold product distribution of Q on $X^{n}$. If $\psi_{n}:X^{n}\to \left(0,\infty \right)$ is such that*

(8)
$$\psi_{n}\left(x^{n}\right)\leq \frac{c_{n}\cdot Z_{Q_{n},\alpha }}{Q_{n}{\left(x^{n}\right)}^{\alpha }}$$

*for some $c_{n}>0$, then*
(a)
*for ρ≠0, we have*

$$\mathrm{sgn}\left(\rho \right)\cdot E_{P_{n}}\left[\psi_{n}{\left(X^{n}\right)}^{\rho }\right]\leq \mathrm{sgn}\left(\rho \right)\cdot 2^{n\rho [H_{\alpha }\left(P\right)+I_{\alpha }\left(P,Q\right)+n^{-1}\log c_{n}]},$$

(b)
*for ρ≠0, we have*

$$\underset{n\to \infty }{lim sup}\frac{1}{n\rho }\log E_{P_{n}}\left[\psi_{n}{\left(X^{n}\right)}^{\rho }\right]\leq H_{\alpha }\left(P\right)+I_{\alpha }\left(P,Q\right)+\underset{n\to \infty }{lim sup}\frac{1}{n}\log c_{n},$$

(c)
*for ρ=0, we have*

$$E_{P_{n}}\left[\log\psi_{n}\left(X^{n}\right)\right]\leq n[H\left(P\right)+I\left(P,Q\right)+n^{-1}\log c_{n}],$$

(d)
*for ρ=0, we have*

$$\underset{n\to \infty }{lim sup}\frac{1}{n}E_{P_{n}}\left[\log\psi_{n}\left(X^{n}\right)\right]\leq H\left(P\right)+I\left(P,Q\right)+\underset{n\to \infty }{lim sup}\frac{1}{n}\log c_{n}.$$




 **Proof.** *Part (a):* From (Equation 8), we have
(9)$$\begin{array}{cc} \mathrm{sgn}\left(\rho \right)\cdot E_{P_{n}}\left[\psi_{n}{\left(X^{n}\right)}^{\rho }\right] & =\mathrm{sgn}\left(\rho \right)\cdot \sum_{x^{n}\in X^{n}}P_{n}\left(x^{n}\right)\psi_{n}{\left(x^{n}\right)}^{\rho }\\ \; & \leq \mathrm{sgn}\left(\rho \right)\cdot c_{n}^{\rho }Z_{Q_{n},\alpha }^{\rho }\sum_{x^{n}\in X^{n}}P_{n}\left(x^{n}\right)Q_{n}{\left(x^{n}\right)}^{-\alpha \rho }\\ \; & =\mathrm{sgn}\left(\rho \right)\cdot 2^{\rho [H_{\alpha }\left(P_{n}\right)+I_{\alpha }\left(P_{n},Q_{n}\right)+\log c_{n}]}\\ \; & =\mathrm{sgn}\left(\rho \right)\cdot 2^{n\rho [H_{\alpha }\left(P\right)+I_{\alpha }\left(P,Q\right)+n^{-1}\log c_{n}]}, \end{array}$$
where the penultimate equality holds from the definition of $I_{\alpha }$, and the last one holds because $H_{\alpha }\left(P_{n}\right)=nH_{\alpha }\left(P\right)$, $I_{\alpha }\left(P_{n},Q_{n}\right)=nI_{\alpha }\left(P,Q\right)$.*Part (b):* Taking log, dividing throughout by nρ, and then applying lim sup successively on both sides of (Equation 9), the result follows.*Part (c):* When ρ=0, we have α=1 and (Equation 8) becomes $\psi_{n}\left(x^{n}\right)\leq \frac{c_{n}}{Q_{n}\left(x^{n}\right)}$. Hence,
(10)$$\begin{array}{cc} E_{P_{n}}\left[\log\psi_{n}\left(X^{n}\right)\right] & =\sum_{x^{n}\in X^{n}}P_{n}\left(x^{n}\right)\log\psi_{n}\left(x^{n}\right)\\ \; & \leq \log c_{n}+\sum_{x^{n}\in X^{n}}P_{n}\left(x^{n}\right)\log\left(1/Q_{n}\left(x^{n}\right)\right)\\ \; & =\log c_{n}+H\left(P_{n}\right)+I\left(P_{n},Q_{n}\right)\\ \; & =n(H\left(P\right)+I\left(P,Q\right)+n^{-1}\log c_{n}), \end{array}$$
where the last equality holds because $H\left(P_{n}\right)=nH\left(P\right)$, and $I\left(P_{n},Q_{n}\right)=nI\left(P,Q\right)$.*Part (d):* Dividing (Equation 10) throughout by *n*, and taking limsup on both sides, the result follows. □

 **Proposition 4.** 
*Let ρ>−1, α=1/(1+ρ), and Q be a probability distribution on X. For n≥1, let $Q_{n}$ be the n-fold product distribution of Q on $X^{n}$. Suppose $\psi_{n}:X^{n}\to \left(0,\infty \right)$ is such that*

$$\psi_{n}\left(x^{n}\right)\geq \frac{a_{n}\;Z_{Q_{n},\alpha }}{Q_{n}{\left(x^{n}\right)}^{\alpha }}$$

*for some $a_{n}>0$, then*
(a)
*for ρ≠0, we have*

$$\mathrm{sgn}\left(\rho \right)\cdot E_{P_{n}}\left[\psi_{n}{\left(X^{n}\right)}^{\rho }\right]\geq \mathrm{sgn}\left(\rho \right)\cdot 2^{n\rho (H_{\alpha }\left(P\right)+I_{\alpha }\left(P,Q\right)+n^{-1}\log\;a_{n})},$$

(b)
*for ρ≠0, we have*

$$\underset{n\to \infty }{lim inf}\frac{1}{n\rho }\log E_{P_{n}}\left[\psi_{n}{\left(X^{n}\right)}^{\rho }\right]\geq H_{\alpha }\left(P\right)+I_{\alpha }\left(P,Q\right)+\underset{n\to \infty }{lim inf}\frac{1}{n}\log a_{n},$$

(c)
*for ρ=0, we have*

$$E_{P_{n}}\left[\log\psi_{n}\left(X^{n}\right)\right]\geq n(H\left(P\right)+I\left(P,Q\right)+n^{-1}\log a_{n}),$$

(d)
*for ρ=0, we have*

$$\underset{n\to \infty }{lim inf}\frac{1}{n}E_{P_{n}}\left[\log\psi_{n}\left(X^{n}\right)\right]\geq H\left(P\right)+I\left(P,Q\right)+\underset{n\to \infty }{lim inf}\frac{1}{n}\log a_{n}.$$




 **Proof.** Similar to proof of Proposition 3. □

## 3. Problem Statements and Known Results

In this section, we discuss Campbell’s source coding problem, Arıkan’s guessing problem, Huleihel et al.’s memoryless guessing problem, and Bunte–Lapidoth’s tasks partitioning problem. Using the general framework presented in the previous section, we re-establish known results, and present a few new results relating to these problems.

### 3.1. Source Coding Problem

Let *X* be a random variable that assumes values from a finite alphabet set $X=\{a_{1},\ldots ,a_{m}\}$ according to a probability distribution *P*. The tuple (X,P) is usually referred to as a *source*. A *binary codeC* is a mapping from X to the set of finite length binary strings. Let L(C(X)) be the length of code C(X). The objective is to find a *uniquely decodable* code that minimizes the expected code-length, that is,
$$\mathrm{Minimize}\;E_{P}\left[L\left(C\left(X\right)\right)\right]$$
over all uniquely decodable codes *C*. Kraft and McMillan independently proved the following relation between uniquely decodable codes and their code-lengths.

Kraft-McMilan Theorem [26]: *If C a uniquely decodable code, then*
(11)$$\sum_{x\in X}2^{-L(C(x))}\leq 1.$$
*Conversely, given a length sequence that satisfies the above inequality, there exists a uniquely decodable code C with the given length sequence.*

Thus, one can confine the search space for *C* to codes satisfying the Kraft–McMillan inequality (Equation 11).

Theorem 5.3.1 of [26]: *If C is a uniquely decodable code, then $E_{P}\left[L\left(C\left(X\right)\right)\right]\geq H\left(P\right)$.*

 **Proof.** Choose $\psi \left(x\right)=2^{L(C(x))}$, where L(C(x)) is the length of code C(x) assigned to alphabet *x*. Since *C* is uniquely decodable, from (Equation 11), we have $\sum_{x\in X}\psi {\left(x\right)}^{-1}\leq 1$. Now, an application of Proposition 2 with k=1 yields the desired result. □

 **Theorem 1.** 
*Let $X^{n}:=X_{1},\ldots ,X_{n}$ be an i.i.d. sequence from $X^{n}$ following the product distribution $P_{n}\left(X^{n}\right)=\prod_{i=1}^{n}P\left(X_{i}\right)$. Let $Q_{n}\left(X^{n}\right)=\prod_{i=1}^{n}Q\left(X_{i}\right)$, where Q is another probability distribution. Let $C_{n}$ be a code such that $L\left(C_{n}\left(X^{n}\right)\right)=\left\lceil -\log Q_{n}\left(X^{n}\right)\right\rceil$. Then, $C_{n}$ satisfies Kraft–McMillan inequality and*

$$\lim_{n\to \infty }\frac{E_{P_{n}}\left[L\left(C_{n}\left(X^{n}\right)\right)\right]}{n}=H\left(P\right)+I\left(P,Q\right).$$



 **Proof.** Choose $\psi_{n}\left(x^{n}\right)=2^{L(C_{n}\left(x^{n}\right))}$, where $L(C_{n}\left(x^{n}\right)$ is the length of code $C_{n}\left(x^{n}\right)$ assigned to sequence $x^{n}$. Then, we have
$$\begin{array}{c} \psi_{n}\left(x^{n}\right)=2^{L(C_{n}\left(x^{n}\right))}=2^{\left\lceil -\log Q_{n}\left(x^{n}\right)\right\rceil }\leq 2\cdot 2^{-\log Q_{n}\left(x^{n}\right)}=2/Q_{n}\left(x^{n}\right). \end{array}$$
An application of Proposition 3 with $c_{n}=2$ yields ${lim sup}_{n\to \infty }E_{P_{n}}\left[L\left(C_{n}\left(X^{n}\right)\right)\right]/n\leq H\left(P\right)+I\left(P,Q\right)$. Furthermore, we also have
$$\begin{array}{c} \psi_{n}\left(x^{n}\right)=2^{L(C_{n}\left(x^{n}\right))}=2^{\left\lceil -\log Q_{n}\left(x^{n}\right)\right\rceil }\geq 2^{-\log Q_{n}\left(x^{n}\right)}=1/Q_{n}\left(x^{n}\right). \end{array}$$
An application of Proposition 4 with $a_{n}=1$ gives ${lim inf}_{n\to \infty }E_{P_{n}}\left[L\left(C_{n}\left(X^{n}\right)\right)\right]/n\geq H\left(P\right)+I\left(P,Q\right)$. □

### 3.2. Campbell Coding Problem

Campbell’s coding problem is similar to Shannon’s source coding problem except that, instead of minimizing the expected code-length, one is interested in minimizing the normalized cumulant of code lengths, that is,
$$\begin{array}{c} \mathrm{Minimize}\;\frac{1}{\rho }\log E_{P}\left[2^{\rho L(C(X))}\right], \end{array}$$
over all uniquely decodable codes *C*, and ρ>0. This problem was shown to be equivalent to minimizing buffer overflow probability by Humblet in [27]. A lower bound for the normalized cumulants in terms of Rényi entropy was provided by Campbell [4].

Lemma 1 of [4]: *Let C be a uniquely decodable code. Then,*
(12)$$\begin{array}{c} \frac{1}{\rho }\log E_{P_{n}}\left[2^{\rho L(C(X))}\right]\geq H_{\alpha }\left(P\right), \end{array}$$
*where α=1/(1+ρ).*

 **Proof.** Apply Proposition 1 with $\psi \left(x\right)=2^{L(C(x))}$ and k=1. □

Notice that, if we ignore the integer constraint of the length function, then
(13)$$\begin{array}{c} L\left(C\left(x\right)\right)=\log\frac{Z_{P,\alpha }}{P{\left(x\right)}^{\alpha }}, \end{array}$$
with $Z_{P,\alpha }$ as in Proposition 1, satisfies (Equation 11) and achieves the lower bound in (Equation 12). Campbell also showed that the lower bound in (Equation 12) can be achieved by encoding long sequences of symbols with code-lengths close to (Equation 13).

Theorem 1 of [4]: *If $C_{n}$ is a uniquely decodable code such that*
(14)$$\begin{array}{c} L\left(C_{n}\left(x^{n}\right)\right)=\left\lceil \log\frac{Z_{P_{n},\alpha }}{P_{n}{\left(x^{n}\right)}^{\alpha }}\right\rceil , \end{array}$$


*then*

$$\lim_{n\to \infty }\frac{1}{n\rho }\log E_{P_{n}}\left[2^{\rho L(C_{n}\left(X^{n}\right))}\right]=H_{\alpha }\left(P\right).$$



 **Proof.** Choose $\psi_{n}\left(x^{n}\right)=2^{L(C_{n}\left(x^{n}\right))}$. Then, from (Equation 14), we have
$$\frac{Z_{P_{n},\alpha }}{P_{n}{\left(x^{n}\right)}^{\alpha }}\leq \psi_{n}\left(x^{n}\right)<2\cdot \frac{Z_{P_{n},\alpha }}{P_{n}{\left(x^{n}\right)}^{\alpha }}\cdot$$The result follows by applying Propositions 3 and 4 with $c_{n}=2$, $a_{n}=1$ and Q=P. □

**Mismatch Case**:

Redundancy in the mismatched case of the Campbell’s problem was studied in [5,6]. Sundaresan showed that the difference in the normalized cumulant from the minimum when encoding according to an arbitrary uniquely decodable code is measured by $I_{\alpha }$-divergence up-to a factor of 1 [5]. We provide a more general version of this result in the following.

 **Proposition 5.** 
*Let X be a random variable that assumes values from set X according to a probability distribution P. Let ρ∈(−1,0)∪(0,∞) and $L:X\to Z_{+}$ be an arbitrary length function that satisfies (Equation 11). Define*

(15)
$$\begin{array}{c} R_{c}\left(P,L,\rho \right):=\frac{1}{\rho }\log E_{P}\left[2^{\rho L(X)}\right]-\min_{K}\frac{1}{\rho }\log E_{P}\left[2^{\rho K(X)}\right], \end{array}$$

*where the minimum is over all length functions K satisfying (Equation 11). Then, there exists a probability distribution $Q_{L}$ such that*

(16)
$$\begin{array}{c} I_{\alpha }\left(P,Q_{L}\right)-\log\eta -1\leq R_{c}\left(P,L,\rho \right)\leq I_{\alpha }\left(P,Q_{L}\right)-\log\eta , \end{array}$$

*where $\eta =\sum_{x}2^{-L(x)}$.*


 **Proof.** Since *K* satisfies (Equation 11), an application of Proposition 1 with $\psi \left(x\right)=2^{K(x)}$ gives us $\frac{1}{\rho }\log E_{P}\left[2^{\rho K(X)}\right]\geq H_{\alpha }\left(P\right)$. Since $K\left(x\right)=\lceil \log(Z_{P,\alpha }/P{\left(x\right)}^{\alpha })\rceil$ satisfies (Equation 11) and $\psi \left(x\right)=2^{K(x)}<2\cdot Z_{P,\alpha }/P{\left(x\right)}^{\alpha }$, applying Proposition 3 with $n=1,c_{1}=2$, and Q=P, we have
$$\frac{1}{\rho }\log E_{P}\left[2^{\rho K(X)}\right]\leq H_{\alpha }\left(P\right)+1,$$
that is, the minimum in (Equation 15) is between $H_{\alpha }\left(P\right)$ and $H_{\alpha }\left(P\right)+1$. Hence,
(17)$$\begin{array}{c} \frac{1}{\rho }\log E_{P}\left[2^{\rho L(X)}\right]-H_{\alpha }\left(P\right)-1\leq R_{c}\left(P,L,\rho \right)\leq \frac{1}{\rho }\log E_{P}\left[2^{\rho L(X)}\right]-H_{\alpha }\left(P\right). \end{array}$$Let us now define a probability distribution $Q_{L}$ as
$$Q_{L}\left(x\right)=\frac{2^{-L(x)/\alpha }}{\sum_{x^{\prime }}2^{-L(x^{\prime })/\alpha }}\cdot$$Then,
$$2^{L(x)}=Q_{L}{\left(x\right)}^{-\alpha }Z_{Q_{L},\alpha }\cdot \frac{1}{\sum_{x^{\prime }}2^{-L(x^{\prime })}}=Q_{L}{\left(x\right)}^{-\alpha }Z_{Q_{L},\alpha }\cdot \frac{1}{\eta },$$
where $\eta =\sum_{x^{\prime }}2^{-L(x^{\prime })}$. Applying Propositions 3 and 4 with n=1, $\psi_{1}\left(x\right)=2^{L(x)}$, $a_{1}=c_{1}=1/\eta$, we obtain
(18)$$\frac{1}{\rho }\log E_{P}\left[2^{\rho L(X)}\right]=H_{\alpha }\left(P\right)+I_{\alpha }\left(P,Q_{L}\right)-\log\eta .$$
Combining (Equation 17) and (Equation 18), we obtain the desired result. □

We remark that the bound in (Equation 16) can be loose when η is small. For example, for a source with two symbols, say *x* and *y*, with code lengths L(x)=L(y)=100, we have $R_{c}\left(P,L,\rho \right)\geq I_{\alpha }\left(P,Q_{L}\right)+98$. However, if one imposes the constraint 1/2≤η≤1, then (Equation 16) simplifies to
$$|R_{c}\left(P,L,\rho \right)-I_{\alpha }\left(P,Q_{L}\right)|\leq 1,$$
which is (Theorem 8 of [5]). $I_{\alpha }\left(P,Q_{L}\right)$ is, in a sense, the penalty when $Q_{L}$ does not match the true distribution *P*. In view of this, a result analogous to Proposition 5 also holds for the Shannon source coding problem.

### 3.3. Arıkan’s Guessing Problem

Let X be a set of objects with |X|=m. Bob thinks of an object *X* (a random variable) from X according to a probability distribution *P*. Alice guesses it by asking questions of the form “Is X=x?”. The objective is to minimize average number of guesses required for Alice to guess *X* correctly. By a guessing strategy (or guessing function), we mean a one-one map G:X→{1,…,m}, where G(x) is to be interpreted as the number of questions required to guess *x* correctly. Arıkan studied the $\rho^{\mathrm{th}}$ moment of number of guesses and found upper and lower bounds in terms of Rényi entropy.

Theorem 1 of [13]: *Let G be any guessing function. Then, for ρ∈(−1,0)∪(0,∞),*
$$\frac{1}{\rho }\log E_{P}\left[G{\left(X\right)}^{\rho }\right]\geq H_{\alpha }\left(P\right)-\log\left(1+\ln m\right).$$

 **Proof.** Let *G* be any guessing function. Let ψ(x)=G(x). Then, we have $\sum_{x\in X}\psi {\left(x\right)}^{-1}=\sum_{x\in X}1/G(x)=\sum_{i=1}^{m}1/i\leq 1+\ln m$. An application of Proposition 1 with k=1+lnm yields the desired result. □

Arıkan showed that an optimal guessing function guesses according to the decreasing order of *P*-probabilities with ties broken using an arbitrary but fixed rule [13]. He also showed that normalized cumulant of an optimal guessing function is bounded above by the Rényi entropy. Next, we present a proof of this using our general framework.

Proposition 4 of [13]: *If $G^{*}$ is an optimal guessing function, then, for ρ∈(−1,0)∪(0,∞),*
$$\frac{1}{\rho }\log E_{P}\left[G^{*}{\left(X\right)}^{\rho }\right]\leq H_{\alpha }\left(P\right).$$

 **Proof.** Let us rearrange the probabilities {P(x),x∈X} in non-increasing order, say
$$p_{1}\geq p_{2}\geq \cdots \geq p_{m}.$$
Then, the optimal guessing function $G^{*}$ is given by $G^{*}\left(x\right)=i$ if $P\left(x\right)=p_{i}$. Let us index the elements in set X as $\left\{x_{1},x_{2},\ldots ,x_{m}\right\}$, according to the decreasing order of their probabilities. Then, for i∈{1,⋯,m}, we have
(19)$$\frac{Z_{P,\alpha }}{P{\left(x_{i}\right)}^{\alpha }}=\frac{\sum_{j=1}^{m}p_{j}^{\alpha }}{p_{i}^{\alpha }}\geq i=G^{*}\left(x_{i}\right).$$
That is, $G^{*}\left(x\right)\leq \frac{Z_{P,\alpha }}{P{\left(x\right)}^{\alpha }}$ for x∈X. Now, an application of Proposition 3 with n=1, Q=P, $\psi_{1}\left(x\right)=G^{*}\left(x\right)$ and $c_{1}=1$, gives us
$$\frac{1}{\rho }\log E_{P}\left[G^{*}{\left(X\right)}^{\rho }\right]=\frac{1}{\rho }\log E_{P}\left[\psi {\left(X\right)}^{\rho }\right]\leq H_{\alpha }\left(P\right)+I_{\alpha }\left(P,P\right)+\log1=H_{\alpha }\left(P\right).$$□

Arıkan also proved that the upper bound of Rényi entropy can be achieved by guessing long sequences of letters in an i.i.d. fashion.

Proposition 5 of [13] *Let $X_{1},X_{2},\ldots ,X_{n}$ be a sequence of i.i.d. guesses. Let $G_{n}^{*}\left(X_{1},\ldots ,X_{n}\right)$ be an optimal guessing function. Then, for ρ∈(−1,0)∪(0,∞),*
$$\lim_{n\to \infty }\frac{1}{n\rho }\log E_{P_{n}}\left[G_{n}^{*}{\left(X_{1},X_{2},\ldots ,X_{n}\right)}^{\rho }\right]=H_{\alpha }\left(P\right).$$

 **Proof.** Let $G_{n}^{*}$ be the optimal guessing function from $X^{n}$ to $\left\{1,2,\ldots ,m^{n}\right\}$. An application of Corollary 1 with $\psi_{n}\left(x^{n}\right)=G_{n}^{*}\left(x^{n}\right)$ and $k_{n}=1+n\ln m$ yields
(20)$$\begin{array}{c} \underset{n\to \infty }{lim inf}\frac{1}{n\rho }\log E_{P_{n}}\left[G_{n}^{*}{\left(X^{n}\right)}^{\rho }\right]\geq H_{\alpha }\left(P\right)-\underset{n\to \infty }{lim sup}\frac{\log(1+n\ln m)}{n}=H_{\alpha }\left(P\right). \end{array}$$
As in the proof of the previous result, we know that $G^{*}\left(x^{n}\right)\leq \frac{Z_{P_{n},\alpha }}{P_{n}{\left(x^{n}\right)}^{\alpha }}$ for $x^{n}\in X^{n}$. Hence, an application of Proposition 3 with $Q_{n}=P_{n}$, $\psi_{n}\left(x^{n}\right)=G_{n}^{*}\left(x^{n}\right)$, and $c_{n}=1$ yields
(21)$$\begin{array}{c} \underset{n\to \infty }{lim sup}\frac{1}{n\rho }\log E_{P_{n}}\left[G_{n}^{*}{\left(X^{n}\right)}^{\rho }\right]\leq H_{\alpha }\left(P\right). \end{array}$$
Combining (20) and (21), we obtain the desired result. □

Henceforth, we shall denote the optimal guessing function corresponding to a probability distribution *P* by $G_{P}$.

**Mismatch Case**:

Suppose Alice does not know the true underlying probability distribution *P*, and guesses according to some guessing function *G*. The following proposition tells us that the penalty for deviating from the optimal guessing function can be measured by $I_{\alpha }$-divergence.

 **Proposition 6.** 
*Let G be an arbitrary guessing function. Then, for ρ∈(−1,0)∪(0,∞), there exists a probability distribution $Q_{G}$ on X such that*

$$\frac{1}{\rho }\log E_{P}\left[G{\left(X\right)}^{\rho }\right]\geq H_{\alpha }\left(P\right)+I_{\alpha }\left(P,Q_{G}\right)-\log\left(1+\ln m\right).$$



 **Proof.** Let *G* be a guessing function. Define a probability distribution $Q_{G}$ on X as
(22)$$Q_{G}\left(x\right)=\frac{G{\left(x\right)}^{-1/\alpha }}{\sum_{x^{\prime }\in X}G{\left(x^{\prime }\right)}^{-1/\alpha }}.$$
Then, we have
$$\frac{Z_{Q_{G},\alpha }}{Q_{G}{\left(x\right)}^{\alpha }}=G\left(x\right)\sum_{x^{\prime }\in X}\frac{1}{G(x^{\prime })}\leq G\left(x\right)\cdot \left(1+\ln m\right)\cdot$$
Now, an application of Proposition 4 with n=1, $\psi_{1}\left(x\right)=G\left(x\right)$, and $a_{1}=1/\left(1+\ln m\right)$ yields the desired result. □

A converse result is the following.

Proposition 1 of [5]: *Let $G_{Q}$ be an optimal guessing function associated with Q. Then, for ρ∈(−1,0)∪(0,∞),*
$$\begin{array}{c} \frac{1}{\rho }\log E_{P}\left[G_{Q}{\left(X\right)}^{\rho }\right]\leq H_{\alpha }\left(P\right)+I_{\alpha }\left(P,Q\right), \end{array}$$
*where the expectation is with respect to P.*

 **Proof.** Let us rearrange the probabilities (Q(x),x∈X) in non-increasing order, say
$$q_{1}\geq q_{2}\geq \cdots \geq q_{m}.$$By definition, $G_{Q}\left(x\right)=i$ if $Q\left(x\right)=q_{i}$. Then, as in (Equation 19), we have $G_{Q}\left(x\right)\leq Z_{Q,\alpha }/Q{\left(x\right)}^{\alpha }$forx∈X. Hence, an application of Proposition 3 with n=1, $\psi_{1}\left(x\right)=G_{Q}\left(x\right)$, and $c_{1}=1$ proves the result. □

Observe that, given a guessing function *G*, if we apply the above proposition for $Q=Q_{G}$, where $Q_{G}$ is as in (22), then we obtain
$$\begin{array}{c} \frac{1}{\rho }\log E_{P_{n}}\left[G_{Q_{G}}{\left(X\right)}^{\rho }\right]\leq H_{\alpha }\left(P\right)+I_{\alpha }\left(P,Q_{G}\right). \end{array}$$

Thus, the above two propositions can be combined to state the following, which is analogous to Proposition 5 (refer Section 3.2).

Theorem 6 of [5]: *Let G be an arbitrary guessing function and $G_{P}$ be the optimal guessing function for P. For ρ∈(−1,0)∪(0,∞), let*
$$R_{g}\left(P,G,\rho \right):=\frac{1}{\rho }\log E_{P}\left[G{\left(X\right)}^{\rho }\right]-\frac{1}{\rho }\log E_{P}\left[G_{P}{\left(X\right)}^{\rho }\right].$$
*Then, there exists a probability distribution $Q_{G}$ such that*
$$|R_{g}\left(P,G,\rho \right)-I_{\alpha }\left(P,Q_{G}\right)|\leq \log\left(1+\ln m\right).$$

### 3.4. Memoryless Guessing

In memoryless guessing, the setup is similar to that of Arıkan’s guessing problem except that this time the guesser Alice comes up with guesses independent of her previous guesses. Let ${\hat{X}}_{1},{\hat{X}}_{2},\ldots$ be Alice’s sequence of independent guesses according to a distribution $\hat{P}$. The guessing function in this problem is defined as
$$\begin{array}{c} G_{\hat{P}}\left(X\right):=\inf\left\{i\geq 1:{\hat{X}}_{i}=X\right\}, \end{array}$$
that is, the number of guesses until a successful guess. Sundaresan [28], inspired by Arıkan’s result, showed that the minimum expected number of guesses required is $\exp\{H_{\frac{1}{2}}\left(P\right)\}$, and the distribution that achieves this is surprisingly not the underlying distribution *P*, but the “tilted distribution” ${\hat{P}}^{*}\left(x\right):=\sqrt{P(x)}/\sum_{y}\sqrt{P(y)}$.

Unlike in Arıkan’s guessing problem, Huleihel et al. [19] minimized what are called factorial moments, defined for $\rho \in Z_{+}$ as
$$\begin{array}{c} V_{\hat{P},\;\rho }\left(X\right)=\frac{1}{\rho !}\prod_{l=0}^{\rho -1}\left(G_{\hat{P}}\left(X\right)+l\right). \end{array}$$

Huleihel et al. [19] (c.f. [20]) studied the following problem.
$$\begin{array}{c} \mathrm{Minimize}\;E_{P}\left[V_{\hat{P},\;\rho }\left(X\right)\right], \end{array}$$
over all $\hat{P}\in P$, where P is the probability simplex that is, $P=\{{\left(P\left(x\right)\right)}_{x\in X}:P\left(x\right)\geq 0,$$\sum_{x}P\left(x\right)=1\}$. Let ${\hat{P}}^{*}$ be the optimal solution of the above problem.

Theorem 1 of [19]: *For any integer ρ>0, we have*
$$\frac{1}{\rho }\log E_{P}\left[V_{{\hat{P}}^{*},\;\rho }\left(X\right)\right]=H_{\alpha }\left(P\right)$$
*and ${\hat{P}}^{*}\left(x\right)=P{\left(x\right)}^{\alpha }/Z_{P,\alpha }.$*

 **Proof.** From [19], we know that
(23)$$\begin{array}{c} E_{P}\left[V_{\hat{P},\;\rho }\left(X\right)\right]=E_{P}\left[\hat{P}{\left(X\right)}^{-\rho }\right]. \end{array}$$Now, the result follows from Proposition 1 with $\psi \left(x\right)=\hat{P}{\left(x\right)}^{-1}$ and k=1. Indeed, since $\hat{P}$ is a probability distribution, we have $\sum_{x\in X}\psi {\left(x\right)}^{-1}=\sum_{x\in X}\hat{P}\left(x\right)=1$. Hence, $\frac{1}{\rho }\log E_{P}\left[V_{\hat{P},\;\rho }\left(X\right)\right]\geq H_{\alpha }\left(P\right)$, and the lower bound is attained by ${\hat{P}}^{*}\left(x\right)=P{\left(x\right)}^{\alpha }/Z_{P,\alpha }$. □

For a sequence of guesses, the above theorem can be stated in the following way. Let ${\hat{X}}^{n}=({\hat{X}}_{1},\ldots ,{\hat{X}}_{n}$), where ${\hat{X}}_{i}$’s are i.i.d. guesses, drawn from $X^{n}$ with distribution ${\hat{P}}_{n}$—the *n*-fold product distribution of $\hat{P}$ on $X^{n}$. If the true underlying distribution is $P_{n}$, then
$$\begin{array}{c} \lim_{n\to \infty }\frac{1}{n\rho }\log E_{P_{n}}\left[V_{{\hat{P}}_{n}^{*}\;\rho }\left(X^{n}\right)\right]=H_{\alpha }\left(P\right), \end{array}$$
where ${\hat{P}}_{n}^{*}\left(x\right)=P_{n}{\left(x\right)}^{\alpha }/Z_{P_{n},\alpha }$. For the mismatched case, we have the following result.

 **Proposition 7.** 
*If the true underlying probability distribution is P, but Alice assumes it as Q and guesses according to its optimal one, namely ${\hat{Q}}^{*}\left(x\right)=Q{\left(x\right)}^{\alpha }/Z_{Q,\alpha }$, then*

$$\frac{1}{\rho }\log E_{P}\left[V_{{\hat{Q}}^{*},\;\rho }\left(X\right)\right]=H_{\alpha }\left(P\right)+I_{\alpha }\left(P,Q\right).$$



 **Proof.** Due to (23), the result follows easily by taking n=1, $\psi_{1}\left(x\right)={\hat{Q}}^{*}{\left(x\right)}^{-1}$, $c_{1}=1,$
$a_{1}=1$ in Propositions 3 and 4. □

### 3.5. Tasks Partitioning Problem

*Encoding of Tasks* problem studied by Bunte and Lapidoth [8] can be phrased in the following way. Let X be a finite set of tasks. A task *X* is randomly drawn from X according to a probability distribution *P*, which may correspond to the frequency of occurrences of tasks. Suppose these tasks are associated with *M* keys. Typically, M<|X|. Due to a limited availability of keys, more than one task may be associated with a single key. When a task needs to be performed, the key associated with it is pressed. Consequently, all tasks associated with this key will be performed. The objective in this problem is to minimize the number of redundant tasks performed. Usual coding techniques suggest assigning tasks with high probability to individual keys and leaving the low probability tasks unassigned. It may just be the case that some tasks may have a higher frequency of occurrence than others. However, for an individual, all tasks can be equally important. If M≥|X|, then one can perform tasks without any redundancy. However, Bunte and Lapidoth [8] showed that, even when M<|X|, one can accomplish the tasks with much less redundancy on average, provided the underlying probability distribution is different from the uniform distribution.

Let $A=\{A_{1},A_{2},\ldots ,A_{M}\}$ be a partition of X that corresponds to the assignment of tasks to *M* keys. Let A(x) be the cardinality of the subset containing *x* in the partition. We shall call *A* the *partition function* associated with partition A. We shall assume that ρ>0 throughout this section, though some of the results hold even when ρ∈(−1,0).

Theorem I.1 of [8]: *The following results hold.*

(a)
*For any partition of X of size M with partition function A, we have*

$$\frac{1}{\rho }\log E_{P}\left[A{\left(X\right)}^{\rho }\right]\geq H_{\alpha }\left(P\right)-\log M.$$

(b)
*If M>log|X|+2, then there exists a partition of X of size at most M with partition function A such that*

$$1\leq E_{P}\left[\hat{A}{\left(X\right)}^{\rho }\right]\leq 1+2^{\rho (H_{\alpha }\left(P\right)-\log\tilde{M})},$$

*where*

(24)
$$\begin{array}{c} \tilde{M}:=\left(M-\log|X|-2\right)/4. \end{array}$$



 **Proof.** *Part (a)*: Let ψ(x)=A(x). Then, we have $\sum_{x\in X}\psi {\left(x\right)}^{-1}=\sum_{x\in X}A{\left(x\right)}^{-1}=M$ (Prop. III-1 of [8]). Now, an application of Proposition 1 with k=M gives us the desired result.*Part (b)*: For the proof of this part, we refer to [8]. □

Bunte and Lapidoth also proved the following limit results.

Theorem I.2 of [8]: *Then, for every n≥1, there exists a partition $A_{n}$ of $X^{n}$ of size at most $M^{n}$ with an associated partition function $A_{n}$ such that*
$$\lim_{n\to \infty }E_{P_{n}}\left[A_{n}{\left(X^{n}\right)}^{\rho }\right]=\left\{\begin{array}{cc} 1 & \mathrm{if}\log M>H_{\alpha }\left(P\right)\\ \infty & \mathrm{if}\log M<H_{\alpha }\left(P\right), \end{array}\right.$$
*where $X^{n}:=\left(X_{1},\cdots ,X_{n}\right)$.*

It should be noted that, in a general set-up of the tasks partitioning problem, it is not necessary that the partition size is of the form $M^{n}$; it can be some $M_{n}$ (a function of *n*). Consequently, we have the following result.

 **Proposition 8.** 
*Let $\left\{M_{n}\right\}$ be a sequence of positive integers such that $M_{n}\geq n\log\left|X\right|+3$, and*

$$\gamma :=\lim_{n\to \infty }\frac{\log M_{n}}{n}$$

*exists. Then, there exists a sequence of partitions of $X^{n}$ of size at most $M_{n}$ with partition functions $A_{n}$ such that*
*(a)* 

$$\lim_{n\to \infty }E_{P_{n}}\left[A_{n}{\left(X^{n}\right)}^{\rho }\right]=1\;\mathrm{if}\;\gamma >H_{\alpha }\left(P\right),$$

*(b)* 

$$\lim_{n\to \infty }\frac{1}{n\rho }\log E_{P_{n}}\left[A_{n}{\left(X^{n}\right)}^{\rho }\right]=H_{\alpha }\left(P\right)-\gamma \;\mathrm{if}\;\gamma <H_{\alpha }\left(P\right).$$




 **Proof.** Let
(25)$$\begin{array}{c} {\tilde{M}}_{n}:=(M_{n}-n\log|X|-2)/4. \end{array}$$
We first claim that $\lim_{n\to \infty }\frac{\log{\tilde{M}}_{n}}{n}=\gamma$. Indeed, since $\frac{\log(1/4)}{n}\leq \frac{\log{\tilde{M}}_{n}}{n}<\frac{\log M_{n}}{n}$, when γ=0, we have $\lim_{n\to \infty }\frac{\log{\tilde{M}}_{n}}{n}=0$. On the other hand, when γ>0, we can find an $n_{\gamma }$ such that $M_{n}\geq 2^{\gamma n/2}\;\forall n\geq n_{\gamma }$. Thus, we have $\lim_{n\to \infty }\frac{n}{M_{n}}=0$. Consequently,
$$\lim_{n\to \infty }\frac{\log{\tilde{M}}_{n}}{n}=\lim_{n\to \infty }\frac{\log M_{n}}{n}+\lim_{n\to \infty }\frac{1}{n}\log\left(1-\frac{\left(n\log|X|+2\right)}{M_{n}}\right)-\lim_{n\to \infty }\frac{2}{n}=\gamma .$$
This proves the claim. From Theorem I.1 of [11], for any n≥1 and $M_{n}>n\log\left|X\right|+2$, there exists a partition $A_{n}$ of $X^{n}$ of size at most $M_{n}$ such that the associated partition function $A_{n}$ satisfies
$$\begin{array}{cc} E_{P_{n}}\left[A_{n}{\left(X^{n}\right)}^{\rho }\right] & \leq 1+2^{\rho (H_{\alpha }\left(P_{n}\right)-\log{\tilde{M}}_{n})}=1+2^{n\rho \left(H_{\alpha }\left(P\right)-\frac{\log{\tilde{M}}_{n}}{n}\right)}. \end{array}$$
*Part (a)*: When $\gamma >H_{\alpha }\left(p\right)$, let us choose $\epsilon =(\gamma -H_{\alpha }\left(P\right))/2>0$. Then, there exists an $n_{\epsilon }$ such that $\frac{\log{\tilde{M}}_{n}}{n}\geq \gamma -\epsilon \;\forall n\geq n_{\epsilon }$. Thus, we have
$$\begin{array}{c} E_{P_{n}}\left[A_{n}{\left(X^{n}\right)}^{\rho }\right]\leq 1+2^{n\rho (H_{\alpha }\left(P\right)-\gamma +\epsilon )}=1+2^{-n\rho (\gamma -H_{\alpha }\left(P\right))/2}\;\forall n\geq n_{\epsilon } \end{array}$$
Consequently, ${lim sup}_{n\to \infty }E_{P_{n}}\left[A_{n}{\left(X^{n}\right)}^{\rho }\right]\leq 1$. We also note that $A_{n}\left(x^{n}\right)\geq 1$ for all $x^{n}\in X^{n}$.Thus, ${lim inf}_{n\to \infty }E_{P_{n}}\left[A_{n}{\left(X^{n}\right)}^{\rho }\right]\geq 1$.*Part (b)*: For any ϵ>0, there exists an $n_{\epsilon }$ such that $\frac{\log{\tilde{M}}_{n}}{n}\geq \gamma -\epsilon \;\forall n\geq n_{\epsilon }$. Thus, we have
$$\begin{array}{cc} E_{P_{n}}\left[A_{n}{\left(X^{n}\right)}^{\rho }\right] & \leq 1+2^{n\rho (H_{\alpha }\left(P\right)-\gamma +\epsilon )}\leq 2^{1+n\rho (H_{\alpha }\left(P\right)-\gamma +\epsilon )}\;\forall n\geq n_{\epsilon } \end{array}$$
Hence, we have
$$\underset{n\to \infty }{lim sup}\frac{1}{n\rho }\log E_{P_{n}}\left[A_{n}{\left(X^{n}\right)}^{\rho }\right]\leq H_{\alpha }\left(P\right)-\gamma +\epsilon \;\forall \epsilon >0$$
Furthermore, an invocation of Corollary 1 with $\psi_{n}\left(x^{n}\right)=A_{n}\left(x^{n}\right)$ and $k_{n}=\sum_{x^{n}\in X^{n}}1/A_{n}\left(x^{n}\right)=M_{n}$ gives us
$$\underset{n\to \infty }{lim inf}\frac{1}{n\rho }\log E_{P_{n}}\left[A_{n}{\left(X^{n}\right)}^{\rho }\right]\geq H_{\alpha }\left(P\right)-\underset{n\to \infty }{lim sup}\frac{\log M_{n}}{n}=H_{\alpha }\left(P\right)-\gamma .$$□

 **Remark 1.** 
*It is interesting to note that, when $\gamma <H_{\alpha }\left(P\right)$, in addition to the fact that $\lim_{n\to \infty }E_{P_{n}}$$[A_{n}{\left(X^{n}\right)}^{\rho }]=\infty$, we also have $E_{P_{n}}\left[A_{n}{\left(X^{n}\right)}^{\rho }\right]\approx 2^{n\rho (H_{\alpha }\left(P\right)-\gamma )}$ for large values of n.*


**Mismatch Case**:

Let us now suppose that one does not know the true underlying probability distribution *P*, but arbitrarily partitions X. Then, the penalty due to such a partition can be measured by the $I_{\alpha }$-divergence as stated in the following theorem.

 **Proposition 9.** 
*Let A be a partition of X of size M with partition function A. Then, there exists a probability distribution $Q_{A}$ on X such that*

$$\frac{1}{\rho }\log E_{P}\left[A{\left(X\right)}^{\rho }\right]=H_{\alpha }\left(P\right)+I_{\alpha }\left(P,Q_{A}\right)-\log M.$$



 **Proof.** Define a probability distribution $Q_{A}=\left\{Q_{A}\left(x\right),\;x\in X\right\}$ as
$$Q_{A}\left(x\right):=\frac{A{\left(x\right)}^{-1/\alpha }}{\sum_{x^{\prime }\in X}A{\left(x^{\prime }\right)}^{-1/\alpha }}\cdot$$Then,
$$\frac{Z_{Q_{A},\alpha }}{Q_{A}{\left(x\right)}^{\alpha }}=A\left(x\right)\sum_{x^{\prime }\in X}\frac{1}{A(x^{\prime })}=A\left(x\right)\cdot M,$$
where the last equality follows due to Proposition III.1 of [8]. Rearranging terms, we have $A\left(x\right)=\frac{Z_{Q_{A},\alpha }}{M\cdot Q_{A}{\left(x\right)}^{\alpha }}$. Hence, an application of Propositions 3 and 4 with n=1, $\psi_{1}\left(x\right)=A\left(x\right)$, $c_{1}=1/M$, $a_{1}=1/M$, and $Q=Q_{A}$ yields the desired result. □

A converse result is the following.

 **Proposition 10.** *Let X be a random task from X following distribution P and ρ∈(0,∞). Let Q be another distribution on X. If M>log|X|+2, then there exists a partition $A_{Q}$ (with an associated partition function $A_{Q}$) of X of size at most M such that*$$E_{P}\left[A_{Q}{\left(X\right)}^{\rho }\right]\leq 1+2^{\rho (H_{\alpha }\left(P\right)+I_{\alpha }\left(P,Q\right)-\log\tilde{M})},$$*where $\tilde{M}$ is as in* (24).

 **Proof.** Similar to proof of Theorem I.1 of [8]. □

## 4. Ordered Tasks Partitioning Problem

In Bunte–Lapidoth’s tasks partitioning problem [8], one is interested in the average number of tasks associated with a key. However, in some scenarios, it might be more important to minimize the average number of redundant tasks performed, before the intended task. To achieve this, tasks associated with a key should be performed in a decreasing order of their probabilities. With such a strategy in place, this problem draws parallel with Arıkan’s guessing problem [13].

Let $A=\{A_{1},A_{2},\ldots ,A_{M}\}$ be a partition of X that corresponds to the assignment of tasks to *M* keys. Let N(x) be the number of redundant tasks performed until and including the intended task *x*. We refer to N(·) as the *count function* associated with partition A. We suppress the dependence of *N* on A for the sake of notational convenience. If *X* denotes the intended task, then we are interested in the $\rho^{\mathrm{th}}$ moment of number of tasks performed, that is, $E_{P}\left[N{\left(X\right)}^{\rho }\right]$, where ρ>0.

 **Lemma 1.** 
*For any count function associated with a partition of size M, we have*

(26)
$$\sum_{x\in X}\frac{1}{N(x)}\leq M\left[1+\ln\left(\frac{\left|X\right|}{M}\right)\right].$$



 **Proof.** For a partition $A=\{A_{1},A_{2},\cdots ,A_{M}\}$ of X, observe that
(27)$$\sum_{x\in X}\frac{1}{N(x)}=\left(1+\frac{1}{2}+\cdots +\frac{1}{|A_{1}|}\right)+\cdots +\left(1+\frac{1}{2}+\cdots +\frac{1}{|A_{M}|}\right).$$Since $1+\frac{1}{2}+\cdots +\frac{1}{|A_{k}|}\leq 1+\ln\left|A_{k}\right|$, for any k∈{1,⋯,M}, we have
(28)$$\begin{array}{cc} \sum_{x\in X}\frac{1}{N(x)} & \leq M+\ln(|A_{1}\left|\cdots \right|A_{M}\left|\right)=M[1+\ln(|A_{1}\left|\cdots \right|A_{M}{\left|\right)}^{1/M}]\\ \; & \overset{\left(a\right)}{\leq }M\left[1+\ln\left(\frac{|A_{1}\left|+\cdots +\right|A_{M}|}{M}\right)\right]=M\left[1+\ln\left(\frac{\left|X\right|}{M}\right)\right], \end{array}$$
where (a) follows due to the AM–GM inequality. □

 **Proposition 11.** 
*Let X be a random task from X following distribution P. Then, the following hold:*
(a)
*For any partition of X of size M, we have*

(29)
$$\begin{array}{c} \frac{1}{\rho }\log E_{P}\left[N{\left(X\right)}^{\rho }\right]\geq H_{\alpha }\left(P\right)-\log\left\{M\left[1+\ln\left(\left|X\right|/M\right)\right]\right\} \end{array}$$

(b)*Let M>log|X|+2. Then, there exists a partition of X of size at most M with count function N such that*$$1\leq E_{P}\left[N{\left(X\right)}^{\rho }\right]\leq 1+2^{\rho (H_{\alpha }\left(P\right)-\log\tilde{M})},$$*where $\tilde{M}$ is as in* (24).


 **Proof.** *Part (a):* Applying Proposition 1 with $k=M\left[1+\ln\left(\left|X\right|/M\right)\right]$ and ψ(x)=N(x), we obtain the desired result.*Part (b):* If *A* and *N* are, respectively, the partition and count functions of a partition A, then we have 1≤N(x)≤A(x) for x∈X. Once we observe this, the proof is same as Theorem I.1 (b) of [8]. □

 **Proposition 12.** 
*Let $\left\{M_{n}\right\}$ be a sequence of positive integers such that $M_{n}\geq n\log\left|X\right|+3$, and $\gamma :=\lim_{n\to \infty }\log M_{n}/n$ exists. Then, there exists a sequence of partitions of $X^{n}$ of size at most $M_{n}$ with count functions $N_{n}$ such that*
*(a)* 

$$\lim_{n\to \infty }E_{P_{n}}\left[N_{n}{\left(X^{n}\right)}^{\rho }\right]=1\;\mathrm{if}\;\gamma >H_{\alpha }\left(P\right),$$

*(b)* 

$$\lim_{n\to \infty }\frac{1}{n\rho }\log E_{P_{n}}\left[N_{n}{\left(X^{n}\right)}^{\rho }\right]=H_{\alpha }\left(P\right)-\gamma \;\mathrm{if}\;\gamma <H_{\alpha }\left(P\right).$$




 **Proof.** Similar to proof of Proposition 8. □

**Remark 2.** 
*(a)* *If we choose the trivial partition, namely $A_{n}=\left\{X^{n}\right\}$, then the ordered tasks partitioning problem simplifies to Arıkan’s guessing problem, that is, we have $M_{n}=1$, $N_{n}\left(x^{n}\right)=G_{n}\left(x^{n}\right)$ and* (26) *simplifies to*
$$\sum_{x^{n}\in X^{n}}\frac{1}{G_{n}\left(x^{n}\right)}\leq 1+n\ln\left|X\right|.$$
*Hence, all results pertaining to the Arıken’s guessing problem can be derived from the ordered tasks partitioning problem.*
*(b)* *Structurally, ordered tasks partitioning problem differs from the Bunte–Lapidoth’s problem only due the factor 1+ln(|X|/M) in* (28)*. While this factor matters for one-shot results, for a sequence of i.i.d. tasks, this factor vanishes asymptotically.*


**Mismatch Case**:

Let us now suppose that one does not know the true underlying probability distribution *P*, but arbitrarily partitions X and executes tasks within each subset of this partition in an arbitrary order. Then, the penalty due to such a partition and ordering can be measured by the $I_{\alpha }$-divergence as stated in the following propositions.

 **Proposition 13.** 
*Let A be a partition of X of size M with count function N. Then, there exists a probability distribution $Q_{N}$ on X such that*

$$\frac{1}{\rho }\log E_{P}\left[N{\left(X\right)}^{\rho }\right]\geq H_{\alpha }\left(P\right)+I_{\alpha }\left(P,Q_{N}\right)-\log\left\{M\left[1+\ln\left(\left|X\right|/M\right)\right]\right\}.$$



 **Proof.** Define a probability distribution $Q_{N}=\left\{Q_{N}\left(x\right),\;x\in X\right\}$ as
$$Q_{N}\left(x\right):=\frac{N{\left(x\right)}^{-1/\alpha }}{\sum_{x^{\prime }\in X}N{\left(x^{\prime }\right)}^{-1/\alpha }}\cdot$$
Then, by Lemma 1, we have
$$\begin{array}{ccc} \frac{Z_{Q_{N},\alpha }}{Q_{N}{\left(x\right)}^{\alpha }} & = & N\left(x\right)\sum_{x^{\prime }\in X}\frac{1}{N(x^{\prime })}\leq N\left(x\right)\cdot M\left[1+\ln\left(\left|X\right|/M\right)\right]. \end{array}$$
Now, an application of Proposition 4 with n=1, $\psi_{1}\left(x\right)=N\left(x\right)$, $Q=Q_{N}$, and $a_{1}=1/M[1+\ln(\left|X\right|/M)]$ yields the desired result. □

A converse result is the following.

 **Proposition 14.** *Let X be a random task from X following distribution P. Let Q be another distribution on X. If M>log|X|+2, then there exists partition $A_{Q}$ (with an associated count function $N_{Q}$) of X of size at most M such that*$$E_{P}\left[N_{Q}{\left(X\right)}^{\rho }\right]\leq 1+2^{\rho (H_{\alpha }\left(P\right)+I_{\alpha }\left(P,Q\right)-\log\tilde{M})}\;\mathrm{if}\;\rho \in \left(0,\infty \right),$$*where $\tilde{M}$ is as in* (24).

 **Proof.** Identical to the proof of Proposition 11(b). □

## 5. Operational Connection among the Problems

In this section, we establish an operational relationship among the five problems (refer Figure 1) that we studied in the previous section. The relationship we are interested in is “Does knowing an optimal or asymptotically optimal solution in one problem helps us find the same in another?” In fact, we end up showing that, under suitable conditions, all the five problems form an equivalence class with respect to the above-mentioned relation.

In this section, we assume ρ>0. First, we make the following observations:Among the five problems discussed in the previous section, only Arıkan’s guessing and Huleihel et al.’s memoryless guessing have a unique optimal solution; others only have asymptotically optimal solutions.Optimal solution of Huleihel et al.’s memoryless guessing problem is the α-scaled measure of the underlying probability distribution *P*. Hence, knowledge about the optimal solution of this problem implies knowledge about an optimal (or asymptotically optimal) solution of all other problems.Among the Bunte–Lapidoth’s and ordered tasks problems, an asymptotically optimal solution of one yields that of the other. The partitioning lemma (Prop. III-2 of [8]) is the key result in these two problems, as it guarantees the existence of the asymptotically optimal partitions in both these problems.

### 5.1. Campbell’s Coding and Arıkan’s Guessing

An attempt to find a close relationship between these two problems was made, for example, by Hanawal and Sundaresan (Section II of [17]). Here, we show the equivalence between asymptotically optimal solutions of these two problems.

 **Proposition 15.** 
*An asymptotically optimal solution exists for Campbell’s source coding problem if and only if an asymptotically optimal solution exists for Arıkan’s guessing problem.*


 **Proof.** Let $\left\{G_{n}^{*}\right\}$ be an asymptotically optimal sequence of guessing functions, that is,
$$\lim_{n\to \infty }\frac{1}{n\rho }\log E_{P_{n}}\left[G_{n}^{*}{\left(X^{n}\right)}^{\rho }\right]=H_{\alpha }\left(P\right).$$
Define
(30)$$Q_{G_{n}^{*}}\left(x^{n}\right):=c_{n}^{-1}\cdot {G_{n}^{*}\left(x^{n}\right)}^{-1},$$
where $c_{n}$ is the normalization constant. Notice that
(31)$$\begin{array}{c} c_{n}=\sum_{x^{n}}{G_{n}^{*}\left(x^{n}\right)}^{-1}\leq 1+n\ln\left|X\right|. \end{array}$$
Let us now define
$$L_{G_{n}^{*}}\left(x^{n}\right):=\left\lceil -\log Q_{G_{n}^{*}}\left(x^{n}\right)\right\rceil .$$
Then, by (Proposition 1 of [17]),
$$L_{G_{n}^{*}}\left(x^{n}\right)\leq \log G_{n}^{*}\left(x^{n}\right)+1+\log c_{n}.$$
Hence,
$$\begin{array}{c} 2^{\rho L_{G_{n}^{*}}\left(x^{n}\right)}\leq 2^{\rho }\cdot c_{n}^{\rho }\cdot G_{n}^{*}{\left(x^{n}\right)}^{\rho }\leq 2^{\rho }\cdot {\left(1+n\ln|X|\right)}^{\rho }\cdot G_{n}^{*}{\left(x^{n}\right)}^{\rho }. \end{array}$$
Thus, we have
$$\underset{n\to \infty }{lim sup}\frac{1}{n\rho }\log E_{P_{n}}\left[2^{\rho L_{G_{n}^{*}}\left(X^{n}\right)}\right]\leq \underset{n\to \infty }{lim sup}\frac{1}{n\rho }\log E_{P_{n}}\left[G_{n}^{*}{\left(X^{n}\right)}^{\rho }\right]=H_{\alpha }\left(P\right).$$
We observe that
$$\begin{array}{c} \sum_{x^{n}\in X^{n}}2^{-L_{G_{n}^{*}}\left(x^{n}\right)}=\sum_{x^{n}\in X^{n}}2^{-\lceil -\log Q_{G_{n}^{*}}\left(x^{n}\right)\rceil }\leq \sum_{x^{n}\in X^{n}}2^{\log Q_{G_{n}^{*}}\left(x^{n}\right)}=1. \end{array}$$
Consequently, from (Equation 12), we have
$$\underset{n\to \infty }{lim inf}\frac{1}{n\rho }\log E_{P_{n}}\left[2^{\rho L_{G_{n}^{*}}\left(X^{n}\right)}\right]\geq H_{\alpha }\left(P\right).$$Thus, $\left\{L_{G_{n}^{*}}\right\}$ is an asymptotically optimal sequence of length functions for Campbell’s coding problem.Conversely, given an asymptotically optimal sequence of length functions $\left\{L_{n}^{*}\right\}$ for Campbell’s coding problem, define
$$Q_{L_{n}^{*}}\left(x^{n}\right):=\frac{2^{-L_{n}^{*}\left(x^{n}\right)}}{\sum_{y^{n}}2^{-L_{n}^{*}\left(x^{n}\right)}}.$$
Let $G_{L_{n}^{*}}$ be the guessing function on $X^{n}$ that guesses according to the decreasing order of $Q_{L_{n}^{*}}$-probabilities. Then, by (Proposition 2 of [17]),
$$\log G_{L_{n}^{*}}\left(x^{n}\right)\leq L_{n}^{*}\left(x^{n}\right).$$
Thus,
$$\underset{n\to \infty }{lim sup}\frac{1}{n\rho }\log E_{P_{n}}\left[G_{L_{n}^{*}}{\left(X^{n}\right)}^{\rho }\right]\leq \lim_{n\to \infty }\frac{1}{n\rho }\log E_{P_{n}}\left[2^{\rho L_{n}^{*}\left(X^{n}\right)}\right]=H_{\alpha }\left(P\right).$$
Furthermore, from Theorem 1 of [13], we have
$$\underset{n\to \infty }{lim inf}\frac{1}{n\rho }\log E_{P_{n}}\left[G_{L_{n}^{*}}{\left(X^{n}\right)}^{\rho }\right]\geq H_{\alpha }\left(P\right)-\underset{n\to \infty }{lim sup}\frac{1}{n}\log\left(1+n\ln|X|\right)=H_{\alpha }\left(P\right).$$
This completes the proof. □

### 5.2. Arıkan’s Guessing and Bunte–Lapidoth’s Tasks Partitioning Problem

Bracher et al. found a close connection between Arıkan’s guessing problem and Bunte–Lapidoth’s tasks partitioning problem in the context of distributed storage [29]. In this section, we establish a different relation between these problems.

 **Proposition 16.** 
*An asymptotically optimal solution of Arıkan’s guessing problem gives rise to an asymptotically optimal solution of tasks partitioning problem.*


 **Proof.** Let $\left\{G_{n}^{*}\right\}$ be an asymptotically optimal sequence of guessing functions. Define
$$Q_{G_{n}^{*}}\left(x^{n}\right)=d_{n}^{-1}G_{n}^{*}{\left(x^{n}\right)}^{-1/\alpha },$$
where $d_{n}$ is the normalization constant. Let $A_{G_{n}^{*}}$ be the partition function satisfying $A_{G_{n}^{*}}\left(x^{n}\right)\leq \left\lceil \beta_{n}\cdot Z_{Q_{G_{n}^{*}},\alpha }/Q_{G_{n}^{*}}{\left(x^{n}\right)}^{\alpha }\right\rceil$ guaranteed by (Proposition III-2 of [8]), where
$$\beta_{n}=\frac{2}{M_{n}-n\log\left|X\right|-2}\cdot$$Thus, we have
$$\begin{array}{ccc} A_{G_{n}^{*}}{\left(x^{n}\right)}^{\rho }\leq {\left\lceil \beta_{n}\cdot Z_{Q_{G_{n}^{*}},\alpha }/Q_{G_{n}^{*}}{\left(x^{n}\right)}^{\alpha }\right\rceil }^{\rho } & \overset{\left(a\right)}{\leq } & \lceil \beta_{n}\cdot \left(1+n\ln|X|\right)\cdot G_{n}^{*}\left(x^{n}\right)\rceil^{\rho }\\ & \overset{\left(b\right)}{\leq } & 1+2^{\rho }\beta_{n}^{\rho }\cdot {\left(1+n\ln|X|\right)}^{\rho }\cdot {G_{n}^{*}\left(x^{n}\right)}^{\rho }, \end{array}$$
where (a) holds because $Z_{Q_{G_{n}^{*}},\alpha }=d_{n}^{-\alpha }\sum_{x^{n}\in X^{n}}G_{n}^{*}{\left(x^{n}\right)}^{-1}\leq d_{n}^{-\alpha }\left(1+n\ln|X|\right)$; and (b) hold because ${\left\lceil x\right\rceil }^{\rho }\leq 1+2^{\rho }x^{\rho }$ for x>0. Hence,
$$\begin{array}{ccc} E_{P_{n}}\left[A_{G_{n}^{*}}{\left(X^{n}\right)}^{\rho }\right] & \leq & 1+2^{\rho }\beta_{n}^{\rho }\cdot {\left(1+n\ln|X|\right)}^{\rho }\cdot E_{P_{n}}\left[G_{n}^{*}{\left(X^{n}\right)}^{\rho }\right]\\ & \overset{\left(c\right)}{\leq } & 1+2^{2\rho }{\left(\frac{1+n\ln|X|}{M_{n}-n\log\left|X\right|-2}\right)}^{\rho }\cdot 2^{n\rho H_{\alpha }\left(P\right)}\\ & = & 1+2^{n\rho \left(H_{\alpha }\left(P\right)+\frac{\log(1+n\ln|X|)}{n}-\frac{\log{\tilde{M}}_{n}}{n}\right)}, \end{array}$$
where ${\tilde{M}}_{n}$ is as in (25); and inequality (c) follows from Proposition 4 of [13] proved in Section 3. Thus, if $M_{n}$ is such that $M_{n}\geq n\log\left|X\right|+3$ and if $\gamma :=\lim_{n\to \infty }\left(\log M_{n}\right)/n$ exists and $\gamma >H_{\alpha }\left(P\right)$, then we have
$$\underset{n\to \infty }{lim sup}E_{P_{n}}\left[A_{G_{n}^{*}}{\left(X^{n}\right)}^{\rho }\right]\leq 1.$$
Since $E_{P_{n}}\left[A_{G_{n}^{*}}{\left(X^{n}\right)}^{\rho }\right]\geq 1$, we have ${lim inf}_{n\to \infty }E_{P_{n}}\left[A_{G_{n}^{*}}{\left(X^{n}\right)}^{\rho }\right]\geq 1$. When $\gamma <H_{\alpha }\left(P\right)$, arguing along the lines of proof of Proposition 8(b), it can be shown than
$$\lim_{n\to \infty }\frac{1}{n\rho }\log E_{P_{n}}\left[A_{G_{n}^{*}}{\left(X^{n}\right)}^{\rho }\right]=H_{\alpha }\left(P\right)-\gamma .$$□

Reverse implication of the above result does not hold always due to the additional parameter $M_{n}$ in the tasks partitioning problem. For example, if $M_{n}={\left|X\right|}^{n}$ and $A_{n}\left(x^{n}\right)=1$ for every $x^{n}$, the partition does not provide any information about the underlying distribution. As a consequence, we will not be able to conclude anything about the optimal (or asymptotically optimal) solutions of other problems. However, if $M_{n}$ is such that $\log M_{n}$ increases sub-linearly, then it does help us find asymptotically optimal solutions of other problems.

 **Proposition 17.** 
*An asymptotically optimal sequence of partition functions $\left\{A_{n}\right\}$ with partition sizes $\left\{M_{n}\right\}$ for the tasks partitioning problem gives rise to an asymptotically optimal solution for the guessing problem if $M_{n}\geq n\log\left|X\right|+3$ and $\lim_{n\to \infty }\left(\log M_{n}\right)/n=0.$*


 **Proof.** By hypothesis,
$$\lim_{n\to \infty }\frac{1}{n\rho }\log E_{P_{n}}\left[A_{n}{\left(X^{n}\right)}^{\rho }\right]=H_{\alpha }\left(P\right).$$
For every $A_{n}$, define the probability distribution
$$Q_{A_{n}}\left(x^{n}\right):=c_{n}^{-1}A_{n}{\left(x^{n}\right)}^{-1},$$
where $c_{n}:=\sum_{x^{n}}A_{n}{\left(x^{n}\right)}^{-1}=M_{n}$. Let $G_{A_{n}}^{*}$ be the guessing function that guesses according to the decreasing order of $Q_{A_{n}}$-probabilities. Then, by (Proposition 2 of [17]), we have
$$\begin{array}{cc} G_{A_{n}}^{*}\left(x^{n}\right)\leq & Q_{A_{n}}{\left(x^{n}\right)}^{-1}=c_{n}A_{n}\left(x^{n}\right)=M_{n}A_{n}\left(x^{n}\right). \end{array}$$
Hence,
$$\underset{n\to \infty }{lim sup}\frac{1}{n\rho }\log E_{P_{n}}\left[G_{A_{n}}^{*}{\left(X^{n}\right)}^{\rho }\right]\leq \underset{n\to \infty }{lim sup}\frac{1}{n}\log M_{n}+\underset{n\to \infty }{lim sup}\frac{1}{n\rho }\log E_{P_{n}}\left[A_{n}{\left(X^{n}\right)}^{\rho }\right]=H_{\alpha }\left(P\right).$$
Furthermore, an application of Theorem 1 of [13] gives us
$$\underset{n\to \infty }{lim inf}\frac{1}{n\rho }\log E_{P_{n}}\left[G_{A_{n}}^{*}{\left(X^{n}\right)}^{\rho }\right]\geq H_{\alpha }\left(P\right)-\underset{n\to \infty }{lim sup}\frac{1}{n}\log\left(1+n\ln|X|\right)=H_{\alpha }\left(P\right).$$
This completes the proof. □

### 5.3. Huleihel et al.’s Memoryless Guessing and Campbell’s Coding

We already know that, if one knows the optimal solution of Huleihel et al.’s memoryless guessing problem, that is, the α-scaled measure of the underlying probability distribution *P*, one has knowledge about the optimal (or asymptotically optimal) solution of Campbell’s coding problem. In this section, we prove a converse statement. We first prove the following lemma.

 **Lemma 2.** 
*Let $L_{n}^{*}$ denote the length function corresponding to an optimal solution for Campbell’s coding problem on the alphabet set $X^{n}$ endowed with the product distribution $P_{n}$. Then, $\sum_{x^{n}\in X^{n}}2^{-L_{n}^{*}\left(x^{n}\right)}\geq 1/2$.*


 **Proof.** Suppose $\sum_{x^{n}\in X^{n}}2^{-L_{n}^{*}\left(x^{n}\right)}<1/2$. Then, we must have $L_{n}^{*}\left(x^{n}\right)\geq 2$ for every $x^{n}\in X^{n}$. Define ${\hat{L}}_{n}\left(x^{n}\right):=L_{n}^{*}\left(x^{n}\right)-1$. We observe that $\sum_{x^{n}\in X^{n}}2^{-{\hat{L}}_{n}\left(x^{n}\right)}<1$, that is, the length function ${\hat{L}}_{n}\left(\cdot \right)$ satisfies (Equation 11). Hence, there exists a code $C_{n}$ for $X^{n}$ such that $L\left(C_{n}\left(x^{n}\right)\right)={\hat{L}}_{n}\left(x^{n}\right)$. Then, for ρ>0, we have $\log E_{P_{n}}\left[2^{\rho L_{n}^{*}\left(X^{n}\right)}\right]>\log E_{P_{n}}\left[2^{\rho {\hat{L}}_{n}\left(X^{n}\right)}\right]$—a contradiction. □

 **Proposition 18.** 
*An asymptotically optimal solution for Huleihel et al.’s memoryless guessing problem exists if an asymptotically optimal solution exists for Campbell’s coding problem.*


 **Proof.** Let $\left\{L_{n}^{*},n\geq 1\right\}$ denote a sequence of asymptotically optimal length functions of Campbell’s coding problem, that is,
(32)$$\begin{array}{c} \lim_{n\to \infty }\frac{1}{n\rho }\log E_{P_{n}}\left[2^{\rho L_{n}^{*}\left(X^{n}\right)}\right]=H_{\alpha }\left(P\right). \end{array}$$
Let us define
$$Q_{L_{n}^{*}}\left(x^{n}\right):=\frac{2^{-L_{n}^{*}\left(x^{n}\right)/\alpha }}{\sum_{{\bar{x}}^{n}\in X^{n}}2^{-L_{n}^{*}\left({\bar{x}}^{n}\right)/\alpha }}.$$
Then, we have
$$\begin{array}{cc} I_{\alpha }\left(P_{n},Q_{L_{n}^{*}}\right) & =\frac{\alpha }{1-\alpha }\log\left(\sum_{x^{n}\in X^{n}}P_{n}\left(x^{n}\right){\left[\sum_{{\hat{x}}^{n}\in X^{n}}{\left(\frac{Q_{L_{n}^{*}}\left({\hat{x}}^{n}\right)}{Q_{L_{n}^{*}}\left(x^{n}\right)}\right)}^{\alpha }\right]}^{\frac{1-\alpha }{\alpha }}\right)-H_{\alpha }\left(P_{n}\right)\\ \; & =\frac{\alpha }{1-\alpha }\log\left(\sum_{x^{n}\in X^{n}}P_{n}\left(x^{n}\right){\left[\sum_{{\hat{x}}^{n}\in X^{n}}\frac{2^{-L_{n}^{*}\left({\hat{x}}^{n}\right)}}{2^{-L_{n}^{*}\left(x^{n}\right)}}\right]}^{\rho }\right)-nH_{\alpha }\left(P\right)\\ \; & =\frac{1}{\rho }\log E_{P_{n}}\left[2^{\rho L_{n}^{*}\left(X^{n}\right)}\right]+\log\zeta_{n}-nH_{\alpha }\left(P\right), \end{array}$$
where $\zeta_{n}=\sum_{{\hat{x}}^{n}\in X^{n}}2^{-L_{n}^{*}\left({\hat{x}}^{n}\right)}$. Hence,
$$\begin{array}{c} \lim_{n\to \infty }\frac{1}{n}I_{\alpha }\left(P_{n},Q_{L_{n}^{*}}\right)=\lim_{n\to \infty }\frac{1}{n\rho }\log E_{P_{n}}\left[2^{\rho L_{n}^{*}\left(X^{n}\right)}\right]+\lim_{n\to \infty }\frac{1}{n}\log\zeta_{n}-H_{\alpha }\left(P\right)\overset{\left(a\right)}{=}0, \end{array}$$
where (a) holds because $\zeta_{n}\in \left[1\;/\;2,1\right]$ (refer Lemma 2). If we assume the underlying probability distribution to be $Q_{L_{n}^{*}}$ instead of $P_{n}$, and perform memoryless guessing according to the escort distribution of $Q_{L_{n}^{*}}$, namely ${\hat{Q}}_{n}^{*}\left(x^{n}\right)=Q_{L_{n}^{*}}{\left(x^{n}\right)}^{\alpha }/Z_{Q_{L_{n}^{*}},\alpha }$, due to Proposition 7, we have
$$\lim_{n\to \infty }\frac{1}{n\rho }\log E_{P_{n}}\left[V_{{\hat{Q}}_{n}^{*},\;\rho }\left(X^{n}\right)\right]=H_{\alpha }\left(P\right)+\lim_{n\to \infty }\frac{1}{n}I_{\alpha }\left(P_{n},Q_{L_{n}^{*}}\right)=H_{\alpha }\left(P\right).$$□

## 6. Summary and Conclusions

This paper was motivated by the need to find a unified framework for the problems on source coding, guessing and the tasks partitioning. To that end, we formulated a general moment minimization problem in the IID-lossless case and observed that the optimal value of its objective function is bounded below by Rényi entropy. We then re-established all achievable lower bounds in each of the above-mentioned problems using the generalized framework. It was interesting to note that the optimal solution did not depend on the moment function ψ, but only on the underlying probability distribution *P* and order of the moment ρ (refer Proposition 1). We also presented a unified framework for the mismatched version of the above-mentioned problems. This framework not only led to refinement of the known theorems, but also helped us identify a few new results. We went on to extend the tasks partitioning problem by asking a more practical question and solved it using the unified theory. Finally, we established an equivalence among these problems, in the sense that an asymptotically optimal solution of one problem yields the asymptotically optimal solution of all other problems. Although the relationship between source coding and guessing is well-known in the literature [30,31,32,33], their connection to the tasks partitioning problem, and the connection in the mismatched version of the problems are new.

Our unified framework also has the potential to act as a general tool-set and provide insights for similar problems in information theory. For example, in Section 4, this framework enabled us to solve a more general tasks partitioning problem, namely, the ordered tasks partitioning problem using this framework. The presented that a unified approach can also be extended and explored further in several ways. This includes (a) *Extension to general alphabet set*: The guessing problem was originally studied for countably infinite alphabet set by Massey [12]. Courtade and Verdú have studied the source coding problem for a countably infinite alphabet set with a cumulant generating function of code word lengths as a design criterion [31]. It would be interesting to see if memory-less guessing and tasks partitioning problems can also be formulated for countably infinite alphabet sets and relationships among the problems can be extended. (b) *More general sources*: Relationship between source coding and guessing is very well-known in the literature. Relationship between guessing and source coding in the ‘with distortion’ case for finite alphabet was established by Merhav and Arıkan [14] and for a countably infinite alphabet by Hanawal and Sundaresan [34]. Relationship between Guessing and Campbell’s coding in the universal case was established by Sundaresan [35]. It would be interesting to see if these can be extended to memoryless guessing and tasks partitioning also, possibly in an unified manner. (c) *Applications*: Arıkan showed an application of the guessing problem in a sequential decoding problem [13]. Humblet showed that the cumulant of code-lengths arises in minimizing the probability of buffer overflow in source coding problems [27]. Rezaee et al. [36], Salamatian et al. [20], and Sundaresan [28] show the application of guessing in the security aspect of password protected systems. Our unified framework has the potential to help solve problems that arise in real life situations and fall in this framework.

## Figures and Tables

**Figure 1 entropy-24-01695-f001:**
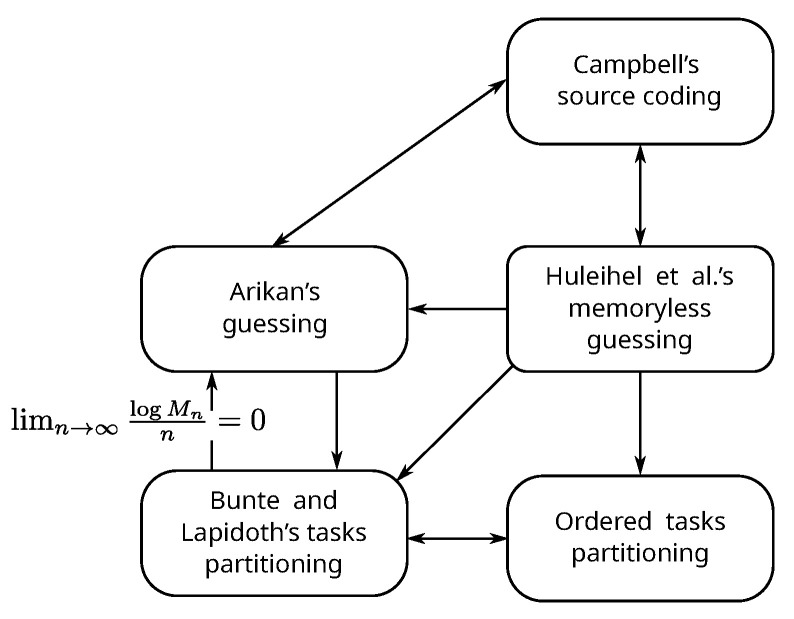
Relationships established among the five problems. A directed arrow from problem *A* to problem *B* means knowing optimal or asymptotically optimal solution of *A* helps us find the same in *B*.

## Data Availability

Not applicable.

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
