# Peer review of "Are Guessing, Source Coding and Tasks Partitioning Birds of A Feather?â€"

_entropy, 2022, doi:10.3390/e24111695_

Round 1
Reviewer 1 Report (New Reviewer)
The paper provides a bridge between Renyi entropy and some practical problems
formulated in terms of information theory.
Introduction and Section 2 clarify terminology and describe the state of art in the area.
Next, the asymptotic bounds on the achievable performance measures are formulated
as theorems.
The paper is well-written and gives insight into the interesting problem.
I recommend publishing the paper as it is.
Author Response
We thank the reviewer for recommending to publish the paper as it is. With due approval from the reviewer, we have changed the title a bit to make sure that the title of this journal version is different from that of the conference one. We hope that the reviewer will be fine with this.
Reviewer 2 Report (New Reviewer)
This is a very interesting and well-written paper. The authors are to be congratulated for submitting such a well-organized and clearly presented manuscript.
The last two lines of the Abstract is not a sentence and is a list of topics that I think was intended to be Key Words. Please correct this oversight. Otherwise, nicely done.
Author Response
In the following we quote the reviewer's comments in italics and our response in usual text.
This is a very interesting and well-written paper. The authors are to be congratulated for submitting such a well-organized and clearly presented manuscript.
We thank the reviewer.
The last two lines of the Abstract is not a sentence and is a list of topics that I think was intended to be Key Words. Please correct this oversight. Otherwise, nicely done.
This has been corrected now. We thank the reviewer.
With due approval from the reviewer, we would also like to inform the reviewer that we have changed the title a bit to make sure that the title of this journal version is different from that of the conference one. We hope that the reviewer will be fine with this.
This manuscript is a resubmission of an earlier submission. The following is a list of the peer review reports and author responses from that submission.
Round 1
Reviewer 1 Report
The authors set out to unify some well-known problems in information theory through a unified lens for both matched and mismatched cases. The paper has some theoretical merit but I believe that is not sufficient for a journal paper publication. What I like about this paper is that it could be a good reference for anyone seeking to understand where Rényi entropies appear in information theory. However, even from that perspective some well known appearances of such quantities are missing from the paper, and some of the claims in the paper are misleading. Let me explain further on these further.
In the matched case, the paper does not really make many new contributions or bring new insights to the individual problems discussed beyond stating some results on each individual problem. In particular, all the results here are known in various forms. Moreover, many of these connections have been observed beyond the moments of these quantities, and for the large deviation performance in [REF3].
In the mismatched case, the literature is more thin. Hence, some of the results in this paper are novel, e.g., I believe Huleihel et al.’s memoryless guessing in the mismatched case is not studied before. On the other hand, the results on the mismatched guesswork do not add much beyond Hanawal and Sundaresan [17]. In particular, it has already been shown by Salamtian et al. [REF1] that the bound in Prop 1 of [32] is loose unless \rho -> 0 (\alpha = 1), and the moments of mismatched guesswork have already been characterized in [REF1].
In Section 5, the paper makes connections between the existing problems. The connection between Campbell’s source coding problem and Massey’s guesswork problem is only true for the value of the moments, and the solutions to the problems are extremely different, even for the zeroth moment we know that one is optimized by Huffman codes and the other is optimized by a coding that mimics the optimal guesswork ordering. In fact, there is a close connection between one-to-one coding and guesswork that has been explored in the literature extensively (see [REF1-4] for the optimal guesswork solutions on how they differ significantly from those of Campbell's source coding). In particular, these problems significantly differ from each other in the mismatched case.
Overall, I remain unconvinced that this paper makes sufficient contributions to the existing body of literature in this space. Besides, the paper needs major revisions to become publishable as it suffers from missing connections and misleading ones. I hope the authors find these comments useful in revising their paper for a future submission.
[REF1] Salamatian, S., Liu, L., Beirami, A. and Médard, M., 2019, August. Mismatched guesswork and one-to-one codes. In 2019 IEEE Information Theory Workshop (ITW) (pp. 1-5). IEEE. available online: https://arxiv.org/abs/1907.00531
[REF2] T. Courtade and S. Verdu, “Cumulant generating function of codeword lengths in optimal lossless compression,” in 2014 IEEE International Symposium on Information Theory (ISIT), July 2014, pp. 2494–2498. Available online: https://people.eecs.berkeley.edu/~courtade/pdfs/CourtadeVerdu_Lossless_ISIT2014.pdf
[REF3] Beirami, A., Calderbank, R., Christiansen, M.M., Duffy, K.R. and Médard, M., 2018. A characterization of guesswork on swiftly tilting curves. IEEE Transactions on Information Theory, 65(5), pp.2850-2871. Available online: https://arxiv.org/abs/1801.09021
[REF4] Kosut, Oliver, and Lalitha Sankar. "Asymptotics and non-asymptotics for universal fixed-to-variable source coding." IEEE Transactions on Information Theory 63, no. 6 (2017): 3757-3772. available online: https://arxiv.org/abs/1412.4444